# Samples of Ba_1−x_Sr_x_Ce_0.9_Y_0.1_O_3−δ_, 0 < x < 0.1, with Improved Chemical Stability in CO_2_-H_2_ Gas-Involving Atmospheres as Potential Electrolytes for a Proton Ceramic Fuel Cell

**DOI:** 10.3390/ma13081874

**Published:** 2020-04-16

**Authors:** Magdalena Dudek, Bartłomiej Lis, Radosław Lach, Salius Daugėla, Tomas Šalkus, Algimantas Kežionis, Michał Mosiałek, Maciej Sitarz, Alicja Rapacz-Kmita, Przemysław Grzywacz

**Affiliations:** 1Faculty of Energy and Fuels, AGH University of Science and Technology, Av. A. Mickiewicza 30, PL-30059 Krakow, Poland; blis@agh.edu.pl (B.L.); grzywacz@agh.edu.pl (P.G.); 2Faculty of Materials Science and Ceramics, AGH University of Science and Technology, Av. A Mickiewicza 30, PL-30059 Krakow, Poland; radek.lach@poczta.fm (R.L.); msitarz@agh.edu.pl (M.S.); kmita@agh.edu.pl (A.R.-K.); 3Faculty of Physics, Institute of Applied Electrodynamics and Telecommunications, Vilnius University, Saulėtekio al. 9/3, LT-10222 Vilnius, Lithuania; daguela.salius@ff.vu.lt (S.D.); algimantas.kezionis@ff.vu.lt (A.K.); 4Jerzy Haber Institute of Catalysis and Surface Chemistry, Polish Academy of Sciences, Niezapominajek 8, PL-30239 Krakow, Poland; tomas.salkus@ff.vu.lt (T.Š.); nbmosial@cyfronet.pl (M.M.)

**Keywords:** high temperature ceramic proton conductors, BaCe_0.9_Y_0.1_O_3_, broadband impedance spectroscopy

## Abstract

Comparative studies were performed on variations in the ABO_3_ perovskite structure, chemical stability in a CO_2_-H_2_ gas atmosphere, and electrical conductivity measurements in air, hydrogen, and humidity-involving gas atmospheres of monophase orthorhombic Ba_1−x_Sr_x_Ce_0.9_Y_0.1_O_3−δ_ samples, where 0 < x < 0.1. The substitution of strontium with barium resulting in Ba_1−x_Sr_x_Ce_0.9_Y_0.1_O_3−δ_ led to an increase in the specific free volume and global instability index when compared to BaCe_0.9_Y_0.1_O_3−δ_. Reductions in the tolerance factor and cell volume were found with increases in the value of x in Ba_1−x_Sr_x_Ce_0.9_Y_0.1_O_3−δ_. Based on the thermogravimetric studies performed for Ba_1−x_Sr_x_Ce_0.9_Y_0.1_O_3−δ_, where 0 < x < 0.1, it was found that modified samples of this type exhibited superior chemical resistance in a CO_2_ gas atmosphere when compared to BaCe_0.9_Y_0.1_O_3−δ_. The application of broadband impedance spectroscopy enabled the determination of the bulk and grain boundary conductivity of Ba_1−x_Sr_x_Ce_0.9_Y_0.1_O_3−δ_ samples within the temperature range 25–730 °C. It was found that Ba_0.98_Sr_0.02_Ce_0.9_Y_0.1_O_3−δ_ exhibited a slightly higher grain interior and grain boundary conductivity when compared to BaCe_0.9_Y_0.1_O_3−δ_. The Ba_0.95_Sr_0.05_Ce_0.9_Y_0.1_O_3−δ_ sample also exhibited improved electrical conductivity in hydrogen gas atmospheres or atmospheres involving humidity. The greater chemical resistance of Ba_1−x_Sr_x_Ce_0.9_Y_0.1_O_3−δ_, where x = 0.02 or 0.05, in a CO_2_ gas atmosphere is desirable for application in proton ceramic fuel cells supplied by rich hydrogen processing gases.

## 1. Introduction

One promising group of ceramic proton conductors is composed of perovskite (ABO_3_)-based oxides. The Y_2_O_3_-doped zirconates BaZr_1−x_Y_x_O_3−δ_ or cerates BaCe_1−x_Y_x_O_3−δ_, where 0 < x < 0.3, are currently considered to be suitable materials for ceramic proton-conducting fuel cells operating at reduced temperatures. Ceramic proton-conducting fuel cells (PCFCs) and solid oxide fuel cells (SOFCs) are being intensively studied with the aim of constructing power sources which can be operated within a temperature range of 400–700 °C [1,2]. Ceramic proton conductors can be applied to the construction of other electrochemical devices important for hydrogen infrastructures. Classic electrochemical devices for this technology include solid oxide electrolyzers, hydrogen or hydrocarbon sensors, hydrogen units for gas purification processes, and reactors for the hydrogenation of compounds [3,4,5]. 

Generally, Y_2_O_3_-doped cerates exhibit higher levels of ionic conductivity than BaZr_1−x_Y_x_O_3−δ_. The limited chemical stability of BaCe_0.9_Y_0.1_O_3−δ_-based materials in H_2_O and CO_2_ gas atmospheres is a crucial factor limiting their practical utilisation in fuel cells supplied by rich hydrogen processing gases. Ba(OH)_2_, BaCO_3_, and CeO_2_ have been observed as secondary phases appearing as a result of the decomposition of yttrium-doped BaCeO_3_ in reaction with CO_2_ [6,7,8,9,10]. Multicomponent solid solutions in which ceria was partially replaced by zirconia in Ba(Ce_1−x_Zr_x_)Y_y_O_3−δ_ exhibited greater resistance to corrosion from CO_2_ attacks. These materials were successfully applied in IT-SOFCs in which power density involving BaCeO_3_-based proton-conducting membranes varied from 30 to 1400 mW/cm^2^ [11,12,13,14,15,16]. 

In the near future, increased use of hydrogen is expected in the fuel-energy sector as well as in various areas of transport. Hydrogen is most often produced from fossil fuels by means of various technologies in which feed streams involving hydrogen, prior to their entry into the PSA cleaning unit, include some impurities, such as CO, CO_2_, N_2_, Ar, and CH_4_. The level of CO_2_ may range from 1.5 to 10 vol.%. Some efforts have also been made to obtain biohydrogen by means of various biological, biotechnological, and renewable energy technologies [17,18,19,20].

The elaboration of either new chemical compositions of ceramic proton-conducting materials, obtained by various processing methods, which are highly tolerant to contamination in hydrogen, or of ways to improve the chemical stability of BaCeO_3_-based electrolytes in a CO_2_ gas atmosphere, is required for the continued development of hydrogen technologies. It was found that the addition of Ba_3_(PO_4_)_2_ or BaWO_4_ to BaCe_0.9_Y_0.1_O_3−δ_ improved resistance to CO_2_ corrosion; however, a deterioration in electrical properties was observed instead [21,22,23]. 

Structural modification via partial substitution of the hosting A ion in ABO_3_ by other ions with the same valence but different ionic radii may also be a way to improve protonic conductivity as well as to reduce chemical reactivity with impurities such as H_2_O or CO_2_ in processing gases. 

ABO_3_ oxides exhibit greater tilts when the A site is characterized by a smaller ion or when the B site is doped with increasing concentrations of M^3+^ (Y, Sm, Gd) dopant [24,25,26]. In the previous paper [27] authors found that Ba_0.95_Ca_0.05_Ce_0.9_Y_0.1_O_3−δ_ exhibited improved corrosion resistance in CO_2_-H_2_ gas atmosphere compared to BaCe_0.9_Y_0.1_O_3−δ_. It was also reported that strontium-modified Ba_0.9_Sr_0.1_Ce_0.8_Y_0.2_O_3−δ_ was characterized by improved chemical stability in H_2_O gas atmospheres compared to BaCe_0.9_Y_0.1_O_3−δ_. The variation in the electrical conductivity of these materials has not been clearly explained [28,29].

The aim of the present study was to determine the effect of the concentration of SrO in Ba_1−x_Sr_x_Ce_0.9_Y_0.1_O_3−δ_, where 0 < x < 0.1, on its electrical and electrochemical properties as well as on its chemical resistance to attacks of CO_2_ corrosion. Another particular goal was the acquisition of a modified ceramic proton conductor characterized by improved chemical stability in a CO_2_ gas atmosphere and by electrical conductivity values comparable to the level of BaCeO_3_-based samples.

## 2. Experimental

### 2.1. The Preparation of Powder and Sintered Samples of Ba_1−x_Sr_x_Ce_0.9_Y_0.1_O_3−δ_, 0 < x < 0.1

Powders from SBCY, i.e., Ba_1−x_Sr_x_Ce_0.9_Y_0.1_O_3−δ_, 0 < x < 0.1, were synthesized via solid-state reaction. The starting reagents were barium carbonate and strontium carbonate (Avantor Performance Materials Poland S.A, Gliwice, Poland, 99.99%), yttrium (III) oxide (Sigma-Aldrich, St.Louis, MO, USA, 99.99%), and cerium(IV) oxide (Acros Organics, Geel, Belgium 99.9%). Stoichiometric amounts of reagents corresponding to the chemical composition of series of samples of BaCe_0.9_Y_0.1_O_3−δ_ (BCY), Ba_0.98_Sr_0.02_Ce_0.9_Y_0.1_O_3−δ_ (2SBCY), Ba_0.95_Sr_0.05_Ce_0.9_Y_0.1_O_3−δ_ (5SBCY) and Ba_0.9_Sr_0.1_Ce_0.9_Y_0.1_O_3−δ_ (10SBCY) were weighed and homogenised. In the SBCY series of samples, the concentration of strontium varied from 2 to 10 mol.%. The powdered chemical reagents were homogenized and milled in a planetary ball mill (Retsch PM 100) in dry ethanol (Avantor Performance Materials Poland S.A, 99.8%). Grinding balls (diameter ~5 mm) made from tetragonal yttria-zirconia (TZP) solid solutions were used in this operation. The milling process elaborated for the 5CBCY sample was applied.

A small portion of mixed reagents prepared for the SBCY-series powders was calcined within a temperature range of 850−1150 °C for 2 h. The monophase orthorhombic phase for all investigated compositions of the SBCY series was found at a temperature of 1150 °C. Subsequently, a 200 g portion of powder was synthesized at 1200 °C for 2 h. The SBCY powders were additionally ground in dry ethanol. The total milling time for monophase SBCY powder samples was 6 h. The details of the homogenisation procedure and milling conditions for the monophase samples was described in the previous paper [30]. The BSCY disc samples were formed via the isostatic pressing method, using a National Forge isostatic cold press, EPSI Engineered Pressure Systems International N.V. (Temse, Belgium). The samples were pressed in a mould with a diameter of 12 under 0.5 MPa. Once pressed, the samples were placed in plastic foil, subjected to outgassing, and pressed under 250 MPa. The samples were sintered at 1550 °C for 2 h in air.

### 2.2. Analytical Methods Used to Evaluate the Physicochemical Properties of the Series of SBCY Samples

X-ray diffraction (XRD) measurements were made using a PANalytical Empyrean diffractometer (Malvern, United Kingdom; CuKα; λ = 0.15418 nm) equipped with a high-temperature measurement oven in order to determine the phase compositions of Ba_1−x_Sr_x_Ce_0.9_Y_0.1_O_3−δ_ powders and sintered samples. XRD patterns were collected in the heating-cooling cycle. The temperatures of XRD measurement varied within a range of 25–800 °C; a ramp of 10 °C min^−1^ was applied in the heating-cooling cycle. Variations in the cell volumes of SBCY samples were calculated based on cell parameters, determined by means of the Rietveld method. In order to find the impact of the incorporation of strontium resulting in Ba_1−x_Sr_x_Ce_0.9_Y_0.1_O_3−δ_ on variations in its structure compared to the BaCe_0.9_Y_0.1_O_3−δ_ sample, the Goldschmidt tolerance factor (t), specific free volume (SFV), and global instability index (GII) were calculated using the SPuDS software package (2.18.06.19, Michael Lufaso, Jacksonville, FL, USA, dedicated to analysis of the structure of ABO_3_-based oxides [27,31,32].

Raman measurements were also carried out for Ba_1−x_SrxCe_0.9_Y_0.1_O_3−δ_ samples. A Horiba Jobin Yvon LabRAM HR micro Raman spectrometer (Higashi-kanda, Chiyoda-ku, Tokyo, Japan) equipped with a CCD detector was used in this investigation. An excitation wavelength of 532 nm was used, with a beam intensity of approximately 10 mW. Acquisition time was set at 30 s [27].

Observations of cross sections as well as surfaces of all SBCY sintered samples were carried out by means of scanning electron microscopy. A Nova NanoSEM 200 (FEI, Eindhoven, Netherlands) coupled with an EDS system was used in these investigations. Comparative observations were made, both of initial SBCY powders, sintered pellets and of samples that had undergone an additional corrosion resistance test in a CO_2_ gas atmosphere [33]. The thermogravimetry (TG) method was used to determine mass variation for the series Ba_1−x_Sr_x_Ce_0.9_Y_0.1_O_3−δ_ in pure CO_2_ gas. A thermobalance (Rubotherm DynTHERM 1100-40 MP-G analyser, TA Instruments, Bochum, Germany) was used in these investigations. TG curves were recorded within a temperature range of 25–1100 °C, at a pressure of 0.1 MPa (abs). Prior to these measurements, all SBCY samples were ground to powder; the powdered SBCY samples were then placed in an alumina crucible. The gas flow was 100 mL min^−1^, the temperature ramp 10 °C min^−1^. The final heating temperature of 1100 °C was maintained for 20 min. Tests of the corrosion resistance of SBCY sintered samples to CO_2_ attack were carried out in a quartz reactor placed in an electric resistance furnace. The SBCY electrolytes were heated to 550 °C and maintained at that temperature for 100 h. A gas mixture containing 2.6 or 5 vol.% of CO_2_ in Ar was used in this experiment. The chemical resistance of BCY and SBCY samples in H_2_O-Ar was examined in analogous conditions. In these experiments, argon was passed first through a scrubber with water, then into the quartz reactor. The SBCY samples used for these experiments were of the same dimensions and thickness as those used in the investigations of electrical properties. Following the experiments, samples were subjected to X-ray investigations and SEM microstructural observations as well as electrical conductivity measurements, performed in both air and hydrogen gas atmospheres [27].

### 2.3. Electrical and Electrochemical Investigations of Ba_1−x_Sr_x_Ce_0.9_Y_0.1_O_3−δ_ Samples

SBCY cylindrical samples with two Pt electrodes prepared using Pt paste (Gwent Electronic Materials Ltd., Taipei, Taiwan) were investigated by means of electrochemical impedance spectroscopy (EIS) in the range 10–1010 Hz with a density of 54 points per decade and an amplitude of 0.1 VRMS in a two-electrode setup [34,35]. EIS measurements were performed in air from room temperature to 730 °C. Electrical conductivity measurements for SBCY samples were also performed in wet 5 vol.% H_2_/Ar gas atmospheres or in wet air. The AC four-probe method was applied. The experiments were carried out within a temperature range of 400–700 °C.

The values of electromotive force (EMF) of the elaborated concentration cells (1):H_2_(I), Pt|SBCY|Pt, H_2_(II)(1)
were used to determine the ionic transport number. Pure humidified hydrogen and a humidified mixture of 10% H_2_ in Ar (flow ~30 mL/min) were used as feeding streams. The experimental procedure was similar to that described in paper [36].

## 3. Results

All of the XRD diffraction patterns recorded for all of the SBCY powders and sintered samples (Figure 1a,b) at RT reflected the BCY monophase orthorhombic Pnma space group. In the case of SBCY samples with increased SrO content in Ba_1−x_Sr_x_Ce_0.9_Y_0.1_O_3−δ_ (Figure 1c), the diffraction peaks shifted towards greater diffraction angles in the recorded XRD patterns. A similar phenomenon was also mentioned by Wang et al. [28]. This suggests that decreases in the cell parameters of SBCY series samples should also be expected. The impact of temperature on possible phase composition changes was also investigated within a temperature range of 25–800 °C.

The variation in the cell volume of Ba_1−x_Sr_x_Ce_0.9_Y_0.1_O_3−δ_ samples vs. chemical composition is shown in Figure 2a. A gradual decrease was observed in cell volume vs. increase in SrO content in the Ba_1−x_Sr_x_Ce_0.9_Y_0.1_O_3−δ_ samples. These results are also the consequences of previously observed shifts in diffraction peaks, in the case of SBCY samples, to angles with greater values in XRD diffraction patterns. Figure 2b presents the representative XRD patterns recorded for a 5SBCY sample during the heating-cooling cycle. No precipitation of impurities as additional phases or phase transitions of the orthorhombic phase of SBCY were found within this temperature range.

Figure 2c presents the dependence of the cell volumes of the BCY and 5SBCY samples during heating-cooling cycles; linear increases were noted. 

The values of the cell volume of 5SBCY, lower than those recorded for BCY, are in close agreement with data reflecting comparison of the ionic radius of the Sr^2+^ ion (r = 135 pm) with the greater radius of the Ba^2+^ ion (r = 149 pm) it replaced. 

Hayashi et al. [37] studied the correlation between structural parameters such as tolerance factor t and specific free volume SFV and variations in values of ionic oxide conductivity for a wide spectrum of ABO_3_-based perovskite oxide-ion conductors. The influence of SFV on oxide-ion migration may have an indirect impact on proton migration as well as on oxide-proton conductivity in ABO_3_-based materials. The authors of the cited paper found that, in order to obtain maximum ionic conductivity as a result of the balance between SFV and t, the optimum value of t was approximately 0.96. It was shown that high values of oxide ion conductivity could be expected for perovskite materials with SFV values higher than 0.4. Neither the structural parameters of Ba_1−x_Sr_x_Ce_0.9_Y_0.1_O_3−δ_ samples nor t, SFV, GII, nor their impact on the samples’ physicochemical properties had been analyzed previously. Figure 3 presents the dependence of calculated values of such parameters as t, SFV, and GII vs chemical composition for Ba_1−x_Sr_x_Ce_0.9_Y_0.1_O_3−δ_ samples.

The highest value of t was found for the BaCe_0.9_Y_0.1_O_3_ sample. The partial replacement of BaO with SrO in Ba_1−x_Sr_x_Ce_0.9_Y_0.1_O_3−δ_ led to a gradual decrease in the tolerance factor vs increased SrO content. This finding is also in close agreement with data reflecting the comparison of the ionic radius of the Sr^2+^ ion (r = 135 pm) with the greater radius of the Ba^2+^ ion (r = 149 pm) which it replaced. On the other hand, introduction of SrO resulting in Ba_1−x_Sr_x_Ce_0.9_Y_0.1_O_3−δ_ led to higher values of SFV. According to data analyzed in the subject literature, an increase in the SFV coefficient is desirable for an improvement in ionic migration. The global instability index (GII) value is typically < 0.1 valence units (vu) for unstrained structures of ABO_3_-based oxides, and as high as 0.2 vu in structures with induced strains. A GII value of approximately 0.2 vu has been found to be critical for several structures typically designated as unstable. The substitution of SrO for BaO in the range 0 < x < 0.1 in Ba_1−x_Sr_x_Ce_0.9_Y_0.1_O_3−δ_ caused a small increase in GII, the values of which were lower than the expected crucial values. 

Possible variations in the ABO_3_ structure in sintered SBCY samples were also investigated, using Raman spectroscopy as complementary to X-ray diffraction analysis. The interpretation of the Raman spectra recorded for BCY and SBCY samples within a range of 400–50 cm^−1^ is presented in Figure 4a–d. It was found that the obtained SBCY samples indicated the same orthorhombic phase, with the Pnma space group.

Due to the similarity of the collected spectra, it can be unequivocally stated that the substitution of SrO for BaO resulting in Ba_1−x_Sr_x_Ce_0.9_Y_0.1_O_3−δ_ was accomplished within the same crystallographic structure. No additional chemical bands in the range 300–100 cm^−1^, which would correspond to variations in ABO_3_ crystal structure, were found in the SBCY series of samples. These findings enabled us to conclude that the introduced Sr^2+^ ions were placed in the same positions as the original Ba^2+^ ions in Ba_1−x_Sr_x_Ce_0.9_Y_0.1_O_3−δ_, 0 < x < 0.1. The crystallographic structure of SBCY sample series was the same as that of BaCe_0.9_Y_0.1_O_3−δ_. The Raman studies performed for SBCY samples were complementary to observations, which were made during the XRD investigations. 

A similar situation was observed for a series of samples of CaO-modified BaCe_0.9_Y_0.1_O_3−δ_ resulting in Ba_0.95_Ca_0.05_Ce_0.9_Y_0.1_O_3−δ_ (5CBCY), whose electrolytic properties were described in the paper [27]. As a result of the analysis of Raman spectra (Figure 4b,c), it can be stated that samples 2SBCY-h and 5SBCY-h, additionally heated in a H_2_O-Ar gas atmosphere, were characterized by greater concentrations of point defects compared to the same samples (2SBCY; 5SBCY) prior to the heating experiments. This may also be attributable to the adsorption of H_2_O from the gas atmosphere and interaction with oxygen vacancies, as evidenced by the significantly lower half-width of the bands on the spectra of the 2SBCY-h and 5SBCY-h samples. This was especially visible in the case of the 2SBCY-h sample [38,39,40]. 

Sintered pellets were prepared from SBCY grounded powders, which were also characterized. Based on XRD investigations (Figure 1a), crystalline size d_(hkl)_ was calculated for all powders from the SBCY series. It was found that the crystalline size d_(211)_ estimated from XRD patterns varied within a range of ~97 to ~106 nm. No significant impact of strontium content in Ba_1−x_Sr_x_Ce_0.9_Y_0.1_O_3−δ_ on the regular variation of crystalline sizes was observed. The SEM/EDS method was also used to investigate ground SBCY powders. The representative SEM recorded for 5SBCY powder is presented in Figure 5a. Based on SEM observation, the particle size falls within the range ~80 to ~400 nm. The agglomerate form of particles was observed in the investigated powders. Figure 5b presents results of chemical analysis performed on the SBCY sample, confirming its established chemical composition.

Figure 6a–d presents the microstructures of the representative surfaces of sintered BCY, 2SBCY, 5SBCY, and 10SBCY samples. All of the sintered SBCY samples were characterised by fairly uniform microstructures and similar levels of closed porosity. 

In the case of sintered BCY samples, the average grain size was ~4 μm. The introduction of SrO into Ba_1−x_Sr_x_(Ce_0.9_Y_0.1_)O_3−δ_ solid solutions led to a slight gradual decrease in average grain size, to 2–3 μm. 

Figure 7a presents relative mass (m/m_o_) variation vs chemical composition recorded for all SBCY powdered samples. The TG curves were recorded in pure CO_2_.

The greatest increase in relative mass (m/m_o_) was determined in the case of the BCY sample. Sawant et al. [41] investigated the chemical stability of BaCeO_3_ as well as of BaCe_0.9_Y_0.1_O_3−δ_ in a CO_2_ gas atmosphere, using the thermogravimetric method within a limited temperature range of 100–900 °C. In these conditions, weight gains were observed above ~450 °C, continuing up to 900 °C due to the following reaction:BaCeO_3_ + CO_2_ → BaCO_3_ + CeO_2_(2)

It was also pointed out that BaCe_0.9_Y_0.1_O_3−δ_ exhibited a higher level of chemical stability than BaCO_3_. In the case of BCY, lesser weight gain was observed in the same conditions. The TG curves we obtained for BCY (Figure 6a) are close to the results of the cited paper [42].

At temperatures from 450 to 900 °C, a continuous increase in mass was observed. No peak of decomposition of carbonate compounds was found on the thermogravimetric curve (m/m_o_ vs temperature) in these conditions. The first mass decrease on the TG curve for BCY was recorded at 970 °C [43]. This may correspond to the thermal decomposition of BaCO_3_. The introduction of SrO into Ba_1−x_Sr_x_Ce_0.9_Y_0.1_O_3−δ_ caused an increase in the maximum temperature decomposition of carbonate compounds in the TG curve. In the case of SBCY, increase in the temperature of carbonate decomposition was accompanied by increased SrO content in Ba_1−x_Sr_x_Ce_0.9_Y_0.1_O_3−δ_ within the temperature range 995–1015 °C. The considerable decrease in mass following attainment of the maximum temperature might be identified as the release of CO_2_ due to the reaction of SrCO_3_/BaCO_3_ with solid solutions of CeO_2_ and Y_2_O_3_, according to reverse reactions (2) and (3).
Ba_(1−x)_Sr_x_Ce_0.9_Y_0.1_O_3(s)_ + CO_2(g)_ → (1–x) BaCO_3(s)_ + x SrCO_3(s)_ + Ce_0.9_Y_0.1_O_1.95(s)_(3)

Two factors were decisive in terms of the probable origin of the faster reaction kinetics of BaCe_0.9_Y_0.1_O_3−δ_ (BCY). One was the basicity of the Ba^2+^ ion, which, being greater than that of the Sr^2+^ ion, led to the greater reactivity of the former, leading in turn to greater chemical reactivity against CO_2_ with acidic properties at the surface of the particle. The second factor may be connected with the more rapid diffusion kinetics in the ABO_3_ crystal structure of Ba^2+^ cations in BaCe_0.9_Y_0.1_O_3−δ_ compared to Sr^2+^ in Ba_1−x_Sr_x_Ce_0.9_Y_0.1_O_3−δ_. Thermogravimetric analysis enabled the observation that the presence of SrO in the Ba_1−x_Sr_x_Ce_0.9_Y_0.1_O_3−δ_ samples caused considerably smaller relative mass changes compared to the BCY sample. In the case of the BCY sample, the increase in relative mass within the investigated temperature range could be attributed to CO_2_ absorption at the surface of the particles and (via bulk diffusion) into the particles. These findings suggested the occurrence of a superficial reaction as well as bulk diffusion, given the bulk diffusion rate of Ba^2+^ for BaCe_0.9_Y_0.1_O_3−δ_, which is higher than that of Sr^2+^ for Ba_1−x_Sr_x_Ce_0.9_Y_0.1_O_3−δ_. This may also be correlated with the structural parameter t and SFV. The least variation in mass was observed for the sample with strontium, where x = 0.02. A larger amount of SrO introduced into Ba_1−x_Sr_x_Ce_0.9_Y_0.1_O_3−δ_ caused a lesser reduction in mass. These results also indicated that the smaller deviation from the original BCY structure may have been responsible for the more rapid kinetics of Ba^2+^ diffusion [44]. 

The chemical stability of the Ba_1−x_Sr_x_Ce_0.9_Y_0.1_O_3−δ_ sintered sample was also tested in gas atmospheres involving 2.6 or 5 vol.% CO_2_ in Ar at 550 °C for 100 h. The XRD diffraction patterns recorded for Ba_1−x_Sr_x_Ce_0.9_Y_0.1_O_3−δ_ samples following additional heating in a 5% Ar-CO_2_ gas atmosphere (Figure 7b) indicated that all samples exhibited precipitation of BaCO_3_ and ceria solid solution. The results obtained from XRD measurements were correlated with data obtained via TG analysis in CO_2_. A quantitative phase analysis performed using the Rietveld method clearly indicated that the amount of precipitation was greater for BCY than for SBCY. The amount of BaCO_3_ for the BCY sample was estimated at about 61%. In the case of SBCY samples, the amount of BaCO_3_ was estimated at 30%–33%. These data clearly indicated that the incorporation of SrO into Ba_1−x_Sr_x_Ce_0.9_Y_0.1_O_3−δ_ inhibited the corrosion of samples in CO_2_ gas atmospheres.

Wang et al. [28] investigated variation in the electrical conductivity of Ba_1−x_Sr_x_Ce_0.8_Y_0.2_O_3−δ_ in wet hydrogen within the temperature range 450–800 °C by means of AC impedance spectroscopy conducted within a frequency range of 0.1 Hz–1 MHz. It was found that electrical conductivity slowly decreased along with increased SrO content in Ba_1−x_Sr_x_Ce_0.8_Y_0.2_O_3−δ_ samples. Greater activation energy of the electrical conductivity process was observed in samples with higher concentrations of SrO in the investigated solid solutions. 

Hung et al. [29] also found that the total electrical conductivity of Ba_1−x_Sr_x_Ce_0.8_Y_0.2_O_3−δ_ samples decreased slightly along with increasing SrO content in air within the temperature range 400–800 °C. Systematic investigations concerning the impact of either grain boundary conductivity or bulk conductivity in Ba_1−x_Sr_x_Ce_0.9_Y_0.1_O_3−δ_ samples are lacking. Data concerning variation in electrical conductivity in hydrogen or humidity-involving atmospheres also require detailed clarification regarding variation in structure and microstructure. The ionic conductivity of BaCeO_3_-based electrolytes is usually measured in air, hydrogen, and steam gas atmospheres using the electrochemical impedance spectroscopy (EIS) method, which is limited to frequencies of 1–10 MHz [45,46,47].

This frequency limit does not reflect all of the phenomena occurring in bulk transport properties at temperatures higher than 450 °C, although this temperature range is crucial for LT- and IT-SOFCs. In an IT-SOFC with a BaCeO_3_-based electrolyte, several processes take place in the bulk, including proton transport, oxygen-ion transport, and, possibly, electronic conductivity. These relaxations at elevated temperatures are shifted well above 10 MHz; therefore, ultrabroadband frequency equipment is necessary to investigate and resolve them [34,35]. 

Ultrabroadband frequency electrochemical impedance spectroscopy within the range 1–10 GHz had previously been applied to the study of the electrical conductivity of BaCe_0.9_Y_0.1_O_3−δ_ and Ba_0.95_Ca_0.05_Ce_0.9_Y_0.1_O_3−δ_ samples [27]. In the present study we continued the application of this method in order to study the electrical properties of SBCY sintered samples. All data were normalised to the sample geometry. The shape of the impedance in the complex plane plots measured in the two-electrode setup of both Ba_0.98_Sr_0.02_Ce_0.9_Y_0.1_O_3−δ_ (2SBCY) and Ba_0.95_Sr_0.05_Ce_0.9_Y_0.1_O_3−δ_ (5SBCY) consisted of semicircles: one separated high-frequency (HF), up to two overlapped medium-frequency (MF), and one separated low-frequency (LF).

Figure 8a–c presents examples of the impedance spectra for 2SBCY and 5SBCY measured at different temperatures.

The bulk and grain boundary ionic conductivity of SBCY sintered samples was determined using equivalent electrical circuits (EECs) consisting of up to four sub-circuits connected in series (Ri, CPEi, where *i* = 1–4). In each sub-circuit, a capacitor was replaced by a constant phase element (CPE) whose impedance is expressed by the following formula:(4)$$Z_{CPE}=\frac{1}{2\pi f_{0}C_{0}}{\left(\frac{f_{0}}{jf}\right)}^{\alpha }$$
where *j* is an imaginary unit, *f* is frequency, *f*_0_ is reference frequency, *C*_0_ is capacitance at reference frequency (1 kHz), and *α* is a coefficient usually assuming a value between 0.5–1.0, close to 1 for an ideal capacitor and close to 0.5 for a diffusion process.

The HF semicircle (R_1_, CPE_1_) was ascribed to grain interior (bulk) processes. The fitted constant phase element (CPE_1_) obtained from the fitting procedure fell within a range from 2 to 18 × 10^–10^ F m^–1^. The first MF semicircle was characterised by capacitances ranging from 3 × 10^–4^ to 8 × 10^–8^ F m^–1^, ascribed to the ionic conductivity of grain boundary regions. The corresponding semicircle was substantially overlapped by the second MF semicircle; thus, the obtained values of grain boundary conductivities are characterised by considerable errors. The other MF and LF semicircles were ascribed to processes occurring in electrodes. 

Figure 8c gives an example comparing the fitted EEC model to measured data. The evolution of the various relaxation processes mentioned above can be inferred from the frequency dependencies of an imaginary impedance part (Figure 9).

The frequency where the maximum of Z″ is observed, called the relaxation frequency, is a good indication for the relaxation dispersion of Z′ (shown in Figure 9a). The relaxation frequency of the charge carrier in the grain interior changes from about 0.2 MHz at 87 °C to about 2 GHz at 707 °C. Consequently, the electrical resistance of grain interiors is characterised by lower values along with increases in temperature. Similarly, the increase in the first MF relaxation frequency, from about 100 Hz to about 40 MHz, was observed over the temperature range shown in Figure 9b. This relaxation frequency was associated with charge carriers in the grain boundaries of electrolyte ceramics. The separation of two MF relaxations was possible only by means of data fitting. The Z″ maximum of the low-frequency relaxation process was observed only at 707 °C. The bulk conductivity dependence vs 1000/T is presented in Figure 10a.

The introduction of strontium resulting in Ba_1−x_Sr_x_Ce_0.9_Y_0.1_O_3−δ_ samples, where 0 < x <0.1, led to variation in the electrical conductivity and activation energy of grain interiors along with SrO content in air within the temperature range 250–1000 K. For all SBCY sintered samples, two areas of variation of bulk conductivity could be distinguished in the plot (σ_B_ – 1000/T). The first region could be observed within the low temperature range, 298–600 K, in which higher activation energy E_a_ was observed. In the temperature range 600–1000 K, lower activation energy of the bulk conductivity of SBCY sinters was observed. Similar dependences of σ_B_ vs temperature were also observed for BaCe_0.9_Y_0.1_O_3−δ_ and Ba_0.95_Ca_0.05_Ce_0.9_Y_0.1_O_3−δ_ sintered samples and for BaCe_1−x_M_x_O_3−δ_, where M = Y, La, Gd and x = 0.1–0.2.

Analysis of electrical conductivity studies vs temperature or partial oxygen pressure indicated that, at temperatures higher than 600 K, oxide-hole conductivity dominates at high oxygen partial pressure (above pO_2_ = 10^–5^ atm) [48,49,50]. Based on analysis of Figure 10a, it can be stated that the 2SBCY sintered sample was characterised by bulk ionic conductivity slightly higher than that of the BCY sample in this condition. Comparative analysis of the chemical resistance vs CO_2_ attack and bulk conductivity of the 2SBCY sample in a CO_2_ gas atmosphere indicated that the introduction of SrO into the Ba_1−x_Sr_x_Ce_0.9_Y_0.1_O_3−δ_ samples (where x = 0.02) led to both an improvement in chemical resistance in a CO_2_ gas atmosphere and an increase in bulk conductivity in comparison to the BCY sample. A continued increase in the share of strontium, i.e., for x > 0.02 in the Ba_1−x_Sr_x_Ce_0.9_Y_0.1_O_3−δ_ samples, led to a gradual decrease in bulk conductivity in air.

Grain boundary conductivity is also an important factor in determining the total electrical conductivity of ceramic material. The high electrical resistance of intergranular areas in a sintered ceramic sample is very often a factor preventing the acquisition of high values of ionic conductivity in proton-conducting electrolytes containing BaZr_0.9_Y_0.1_O_3−δ_ (BZY) or BaCe_0.9_Y_0.1_O_3−δ_ (BCY) [51,52]. Lindman presented an analysis of the potential for reducing the value of resistance of grain boundaries for BZY and BCY. The results of the considerations presented in the cited paper suggest that BaCe_0.9_Y_0.1_O_3−δ_ [53] exhibits a greater tendency towards modification of grain boundaries aimed at reducing their electrical resistance. The variation in grain boundary ionic conductivity vs 1000/T for sintered Ba_1−x_Sr_x_Ce_0.9_Y_0.1_O_3−δ_ samples is presented in Figure 10b. Based on the data presented therein, it can be observed that the gradual substitution of SrO for BaO in Ba_1−x_Sr_x_Ce_0.9_Y_0.1_O_3−δ_ samples, where 0 < x < 0.1, led to slight changes in grain boundary conductivity in relation to the BCY sample. Improved grain boundary conductivity was observed for the 2SBCY sample. It is also widely known that the introduction of small amounts of SrO or CaO into a CeO_2_-Gd_2_O_3_-SrO solution enable the acquisition of greater electrical conductivity as a result of the potential for cleansing the grain boundary area of resistive non-organic compounds [54,55]. In the case of Ba_1−x_Sr_x_Ce_0.9_Y_0.1_O_3−δ_ samples, where x > 0.02, with higher SrO concentrations, a deterioration in grain boundary conductivity was observed.

The total electrical conductivity σ of the sintered Ba_1−x_Sr_x_Ce_0.9_Y_0.1_O_3−δ_ samples within the temperature range 400–800 °C was also measured using AC electrochemical impedance spectroscopy in 5 vol.% H_2_ in Ar. Figure 11a presents variations in σ vs temperature for Ba_1−x_Sr_x_Ce_0.9_Y_0.1_O_3−δ_ sintered samples.

It can be concluded that the sintered 5SBCY sample was characterised by the highest value of electrical conductivity and the lowest value of activation energy in an atmosphere of humidified hydrogen. In the case of Ba_1−x_Sr_x_Ce_0.9_Y_0.1_O_3−δ_ samples, the partial replacement of BaO with SrO led to a small improvement in total conductivity σ in an atmosphere of humidified hydrogen gas. The t_ion_ transference number was also estimated from EMF measurement of the steam/hydrogen concentration cell (1) (Figure 11b). In these conditions, the transference number determined for the BCY sample fell within the range 0.90–0.95. The 5SBCY and 2SBCY sintered samples exhibited values of the t_ion_ transference number similar to that of the BCY electrolyte. These results confirmed that the investigated SBCY samples exhibited the same level of ionic conductivity as the BCY ceramic membrane.

The electrical conductivity of the SBCY samples was also measured following additional heat treatment at 550 °C for 100 h in a CO2/Ar gas atmosphere involving 2.6 or 5 vol.% CO_2_.

Figure 12a presents the impedance spectra (Z″–Z′) recorded for the 5SBCY sample before and after additional heating in a 5 vol.% CO_2_ gas atmosphere at 390 °C. As can be seen, increases in the electrical resistance of grain interiors and grain boundary regions of the 5SBCY sample were observed following additional heat treatment in a CO_2_ gas atmosphere. Figure 12b shows the variation in σ vs 1000/T determined for the 5SBCY sample and presents the electrical conductivity values obtained for samples before and after additional exposure in 5 or 2.6 vol.% CO_2_ in Ar. All measurements were performed via the EIS method. A slight decrease in electrical conductivity was recorded for 5SBCY following additional tests performed in a gas atmosphere involving 2.6 or 5 vol.% CO_2_.

The impact of the presence of H_2_O in an Ar gas atmosphere on the total electrical conductivity of the 5SBCY sample was also studied. In Figure 13, the electrical conductivity values recorded before and after an additional test in 5 vol.% H_2_O in Ar (h) are plotted. The comparative analysis of electrical conductivity data for the 5SBCY electrolyte before and after test enabled us to state that exposure of the sample in an H_2_O gas atmosphere led to a slight improvement in electrical conductivity. This finding may be directly connected with the incorporation of water in the 5SBCY sample.

## 4. Conclusions

All of the investigated Ba_1−x_Sr_x_Ce_0.9_Y_0.1_O_3−δ_ samples were monophase. The orthorhombic (Pnma space group) structure was identified exclusively using the XRD diffraction method. In the case of SBCY samples with increased SrO content in Ba_1−x_Sr_x_Ce_0.9_Y_0.1_O_3−δ_ the diffraction peaks shifted towards greater diffraction angles in the recorded XRD patterns; consequently, a decrease was found in the cell volume of Ba_1−x_Sr_x_Ce_0.9_Y_0.1_O_3−δ_ samples when compared to BaCe_0.9_Y_0.1_O_3−δ_. Raman studies confirmed that the introduced Sr^2+^ ions were located in the same position in the crystallographic structure in Ba_0.98_Sr_0.02_Ce_0.9_Y_0.1_O_3−δ_, Ba_0.95_Sr_0.05_Ce_0.9_Y_0.1_O_3−δ_, and Ba_0.9_Sr_0.1_Ce_0.9_Y_0.1_O_3−δ_ as the original Ba^2+^ ions in BaCe_0.9_Y_0.1_O_3−δ_. The increase in the concentration of SrO in Ba_1−x_Sr_x_Ce_0.9_Y_0.1_O_3−δ_, where 0 < x < 0.1, also led to a decrease in tolerance factor as well as increases in specific free volume and global instability factor. The analyzed structural parameters were responsible for variation in chemical resistance in CO_2_ and electrical conductivity in air, hydrogen, or wet-gas-involving atmospheres. It was found that Ba_1−x_Sr_x_Ce_0.9_Y_0.1_O_3−δ_, where 0.02 < x < 0.05, exhibited improved electrical conductivity in air as well as in hydrogen or humidity-involving gas atmospheres. Superior chemical stability in CO_2_-involving gas atmospheres was also observed in the case of these samples in comparison to BaCe_0.9_Y_0.1_O_3−δ_. The values of the t_ion_ transference number determined in hydrogen concentration galvanic cells for Ba_0.98_Sr_0.02_Ce_0.9_Y_0.1_O_3−δ_ and Ba_0.95_Sr_0.05_Ce_0.9_Y_0.1_O_3−δ_ are comparable to data obtained for BaCe_0.9_Y_0.1_O_3−δ_. The composition of Ba_0.98_Sr_0.02_Ce_0.9_Y_0.1_O_3−δ_ or Ba_0.95_Sr_0.05_Ce_0.9_Y_0.1_O_3−δ_ mentioned above exhibited not only the same level of electrical conductivity as BaCe_0.9_Y_0.1_O_3−δ_, but also improved chemical resistance in 2.6–5.0 vol.% CO_2_ gas atmospheres. The variations in electrical conductivity recorded for the Ba_0.95_Sr_0.05_Ce_0.9_Y_0.1_O_3−δ_ sample following exposure in gas atmospheres with close to 5 vol.% of CO_2_ were smaller than those for BaCe_0.9_Y_0.1_O_3−δ_ and the previously investigated Ba_0.95_Ca_0.05_Ce_0.9_Y_0.1_O_3−δ_. These materials appear to be more suitable ceramic proton-conducting membranes for the construction of ceramic fuel cells operating in hydrogen gas atmospheres involving contamination of CO_2_ below 2 vol.%.

## Figures and Tables

**Figure 1 materials-13-01874-f001:**
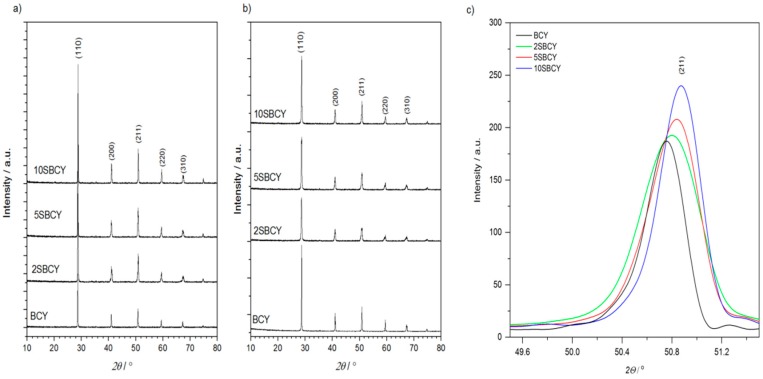
XRD patterns collected for Ba_1−x_Sr_x_Ce_0.9_Y_0.1_O_3−δ_ (0 < x < 0.1) (SBCY) powders (**a**), sintered samples (**b**), and diffraction peak (211) shifts of SBCY samples toward greater diffraction angles (**c**) vs. chemical composition.

**Figure 2 materials-13-01874-f002:**
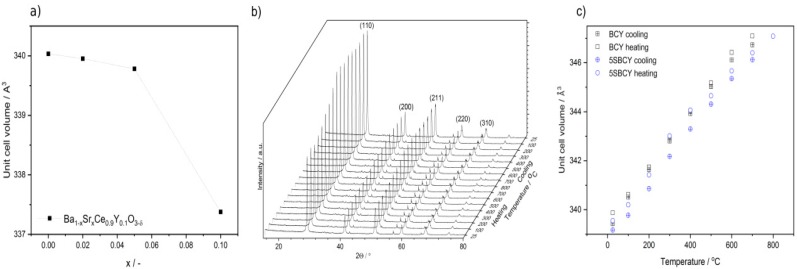
Variation in the cell volume of Ba_1−x_Sr_x_Ce_0.9_Y_0.1_O_3−δ_ vs chemical composition at RT (**a**); XRD patterns collected for Ba_0.95_Sr_0.05_Ce_0.9_Y_0.1_O_3−δ_ (5SBCY) during the heating-cooling cycle within the temperature range 25–800 °C (**b**) variation in BaCe_0.9_Y_0.1_O_3−δ_ (BCY) and 5SBCY cell volume vs temperature (**c**).

**Figure 3 materials-13-01874-f003:**
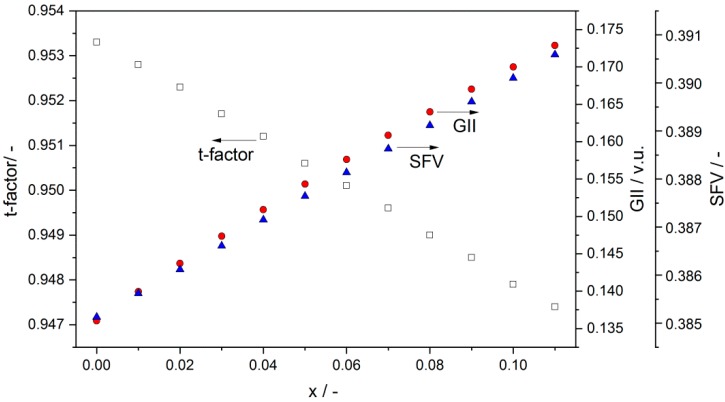
Dependence of the t tolerance factor, specific free volume SFV, and global instability index GII vs strontium content in Ba_1−x_Sr_x_Ce_0.9_Y_0.1_O_3−δ_ samples.

**Figure 4 materials-13-01874-f004:**
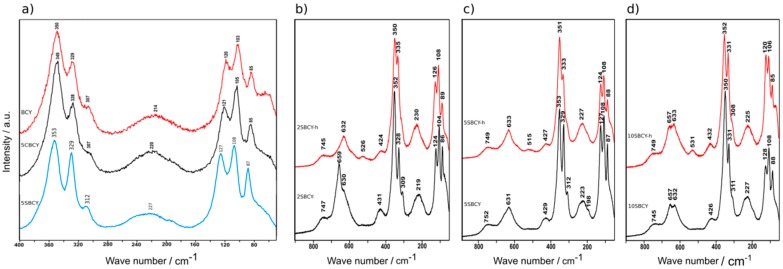
Raman spectra recorded for BCY (**a**), Ba_0.98_Sr_0.02_Ce_0.9_Y_0.1_O_3−δ_ (2SBCY) (**b**), 5SBCY (**c**), and Ba_0.9_Sr_0.1_Ce_0.9_Y_0.1_O_3−δ_ (10SBCY) (**d**) samples before and after additional heating (h) in 5 vol.% H_2_O-Ar at 550 °C for 100 h.

**Figure 5 materials-13-01874-f005:**
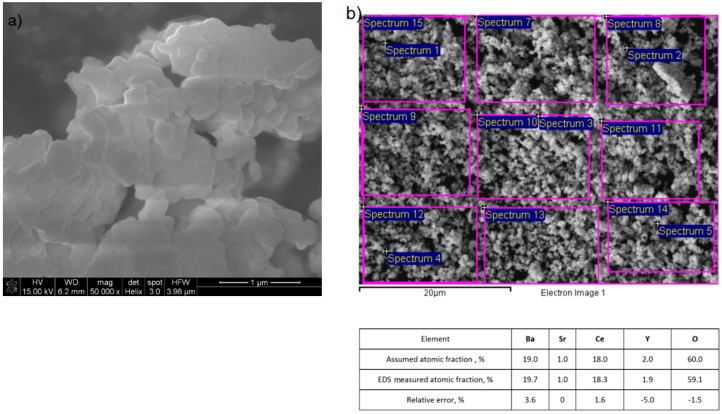
The representative SEM image recorded for 5SBCY powder (**a**) and SEM/EDS chemical analysis (**b**) performed for this sample.

**Figure 6 materials-13-01874-f006:**
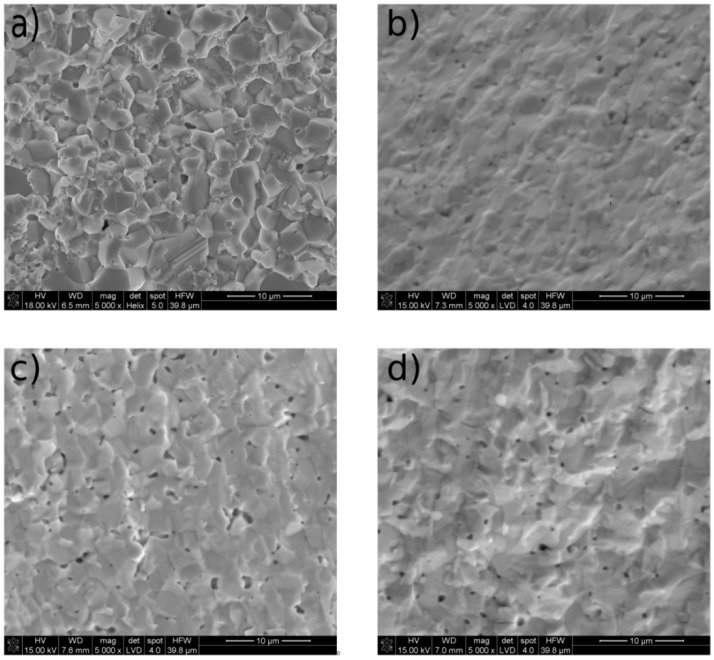
The microstructure of the representative surface of sintered (**a**) BCY, (**b**) 2SBCY, (**c**) 5SBCY, and (**d**) 10SBCY samples.

**Figure 7 materials-13-01874-f007:**
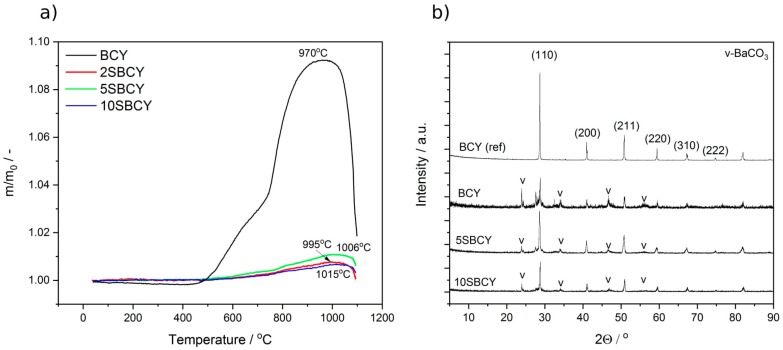
TG curves recorded for Ba_1−x_Sr_x_Ce_0.9_Y_0.1_O_3−δ_ samples in a pure CO_2_ gas atmosphere, at temperatures ranging from 25 to–1100 °C (**a**); XRD pattern recorded for SBCY samples after heating in 5 vol.% CO_2_ in an Ar gas atmosphere at 550 °C for 100 h (**b**).

**Figure 8 materials-13-01874-f008:**
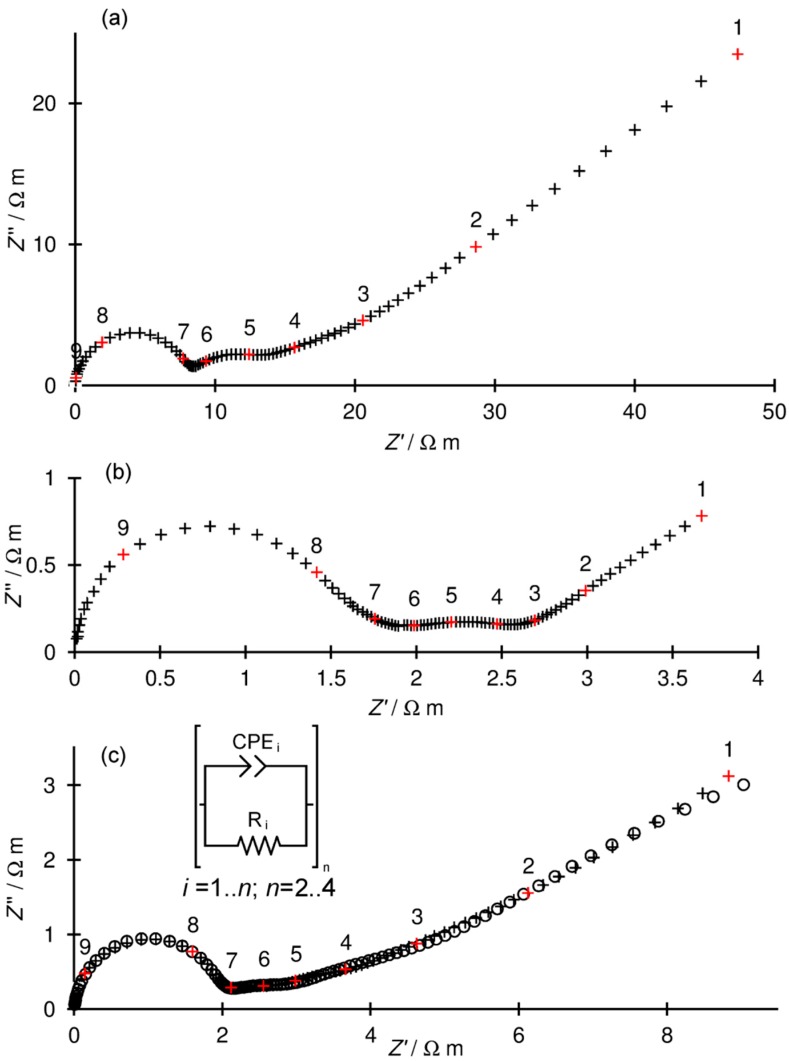
Complex impedance plane plots measured in air (**a**) at 287 °C for 2SBCY ceramics and (**b**) at 507 °C for 5SBCY ceramics; (**c**) an example of fit for 2SBCY ceramics measured at 407 °C; crosses indicate measured data, circles indicate the fitted model; numbers above red symbols denote logarithms of frequency.

**Figure 9 materials-13-01874-f009:**
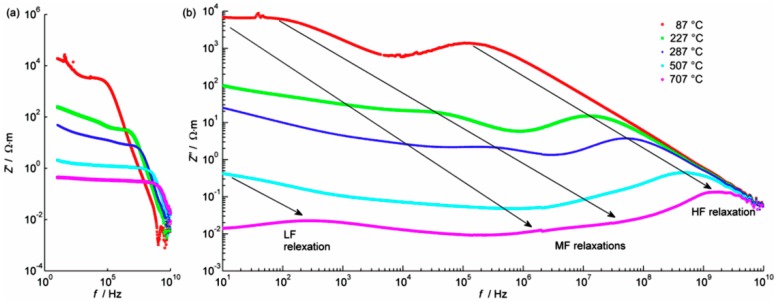
Spectra of (**a**) real Z′ and (**b**) imaginary Z″ parts for a 2SBCY sintered pellet. Arrows indicate (approximately) the shift of the relaxation frequency of several processes with changes in temperature.

**Figure 10 materials-13-01874-f010:**
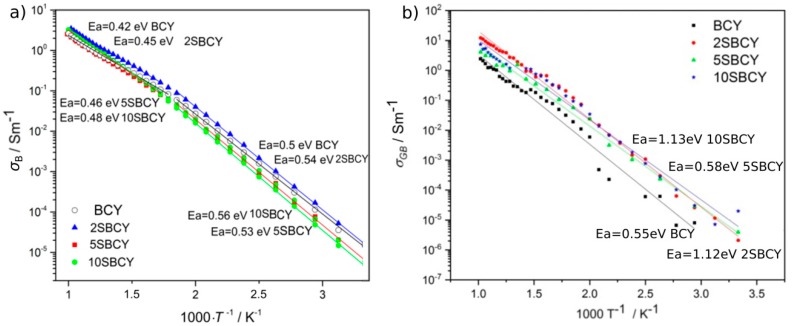
Variation in grain interior σ_B_ (**a**) and grain boundary σ_GB_ (**b**) conductivities of the SBCY sample series vs temperature.

**Figure 11 materials-13-01874-f011:**
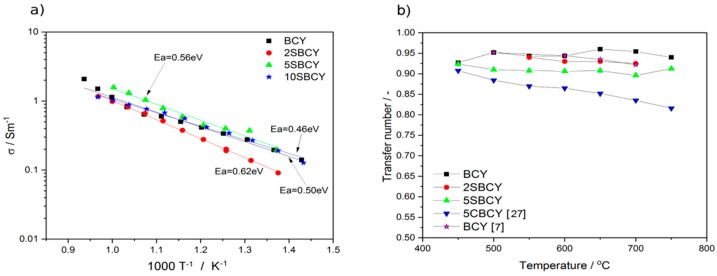
Independence of total electrical conductivity σ for SBCY sample series vs temperature recorded in H_2_O-5 vol.% H_2_ in Ar (**a**); variation in the t_ion_ transference number of BCY and 5SBCY ceramics determined by means of EMF measurements of a hydrogen concentration cell (**b**).

**Figure 12 materials-13-01874-f012:**
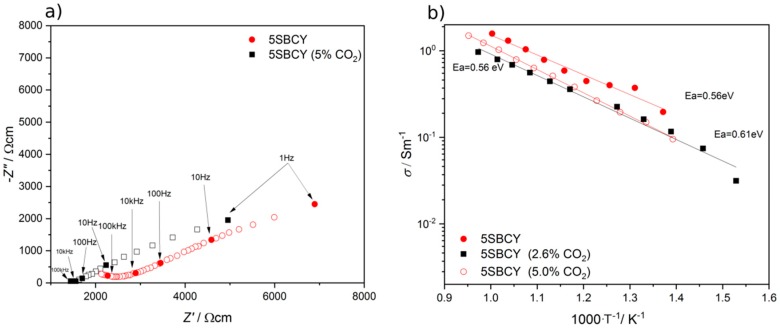
Impedance spectra (Z″–Z′) recorded for the 5SBCY sample before and after additional heating in a 5 vol.% CO_2_ gas atmosphere at 390 °C (**a**); variation in σ vs temperature recorded for 5SBCY before and after additional exposure in 5 or 2.6 vol.% CO_2_ (**b**).

**Figure 13 materials-13-01874-f013:**
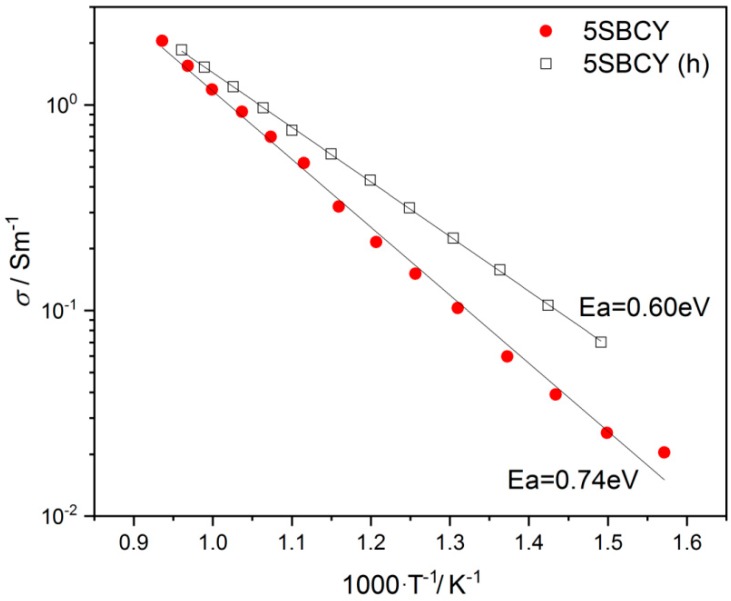
Variation σ vs 1000/T recorded for the 5SBCY sample before or after additional exposure in 5 vol.% H_2_O in Ar (h).

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
