# Peer review of "Samples of Ba1−xSrxCe0.9Y0.1O3−δ, 0 < x < 0.1, with Improved Chemical Stability in CO2-H2 Gas-Involving Atmospheres as Potential Electrolytes for a Proton Ceramic Fuel Cell"

_materials, 2020, doi:10.3390/ma13081874_

Round 1

Reviewer 1 Report

The authors report “Some observations on the ionic conductivity and chemical stability in a CO2 gas atmosphere of samples of Ba1–xSrxCe0.9Y0.1O3, 0 < x < 0.1, as potential electrolytes for proton ceramic fuel cell”. Although the work presented is interesting to semiconductor materials research community, this manuscript should be improved prior to potential publication. Please find the major concerns below.

  • The title is long and not so appealing, the authors should reconsider the title to be precise.
  • Sintering of SBCY was studied with in the temperature range of 25 & 800 °C. Why did the authors utilize that temperature range? What if is the sintering behavior beyond 800 °C ?
  • It is apparent from the characterizations that there is no real measurement of particle size/crystallite size, so the authors should utilize XRD patterns to estimate the crystallite sizes using Scherrer’s formula (see for example, https://doi.org/10.1021/la0477183 and DOI:10.1021/acs.chemmater.6b01749 and cite this relevant literature). It seems there is some change in the crystallite sizes as some diffraction peaks are broader than the other and the authors should explain this.
  • Did the authors observe any XRD diffraction peaks shift due to Sr incorporation into Ba based crystal? Please comment on this in the manuscript. From Raman spectra it is convincing that Sr2+ ions are present in the same place/location of Ba2+, if that is the case XRD diffraction patterns should slightly shift towards higher 2 theta because of larger Ba is being replaced by smaller Sr (for a similar results and discussion please see DOI: 10.1039/c8nr04399j).
  • The authors claiming from that modification of the structure and physicochemical properties observed because of partial replacement of barium with strontium. What is meant by partial replacement here? Is it up to a certain percentage of Ba was replaced by Sr? If so how much? Did the authors try to collect any characterization to prove Ba & Sr elemental composition within the samples? The authors can utilize ICP-OES or at least SEM/EDAX to estimate the elemental compositions and this also helps in further understanding the so-called modifications. Top of Form

Author Response

Dear Reviewer 1,

Thank you very much for reviewing our paper

The Ba1–xSrxCe0.9Y0.1O3, 0 < x < 0.1, with improved chemical stability  in CO2 gas involving atmospheres as potential electrolytes for proton ceramic fuel cell

 Your valuable remarks enabled us to improve the quality of our paper. Now I would like to take this opportunity to respond to all your remarks

The new title

Samples of Ba1–xSrxCe0.9Y0.1O3, 0 < x < 0.1, with improved chemical stability in CO2-H2 gas involving atmospheres as potential electrolytes for a proton ceramic fuel cell

All answers for your question and remarks were included in attached file

The paper was checked by native speaker

Best regards

Magdalena Dudek

Reviewer 2 Report

Manuscript Number: materials-741997

Title: Some observations on the ionic conductivity and chemical stability in a CO2 gas atmosphere of samples of Ba1–xSrxCe0.9Y0.1O3, 0 < x < 0.1, as potential electrolytes for proton ceramic fuel cell

General Comments:

In this work comparative studies of variation in the ABO3 perovskite structure and physicochemical and electrochemical properties of Ba1–xSrxCe0.9Y0.1O3–δ sintered ceramics, where 0 < x < 0.1, are described. It was found that the partial substitution of strontium for barium resulting in Ba1–xSrxCe0.9Y0.1O3–δ caused an increase in specific free volume and global instability index compared to the original composition, i.e. BaCe0.9Y0.1O3–δ. It was also argued that Ba0.98Sr0.02Ce0.9Y0.1O3 exhibits slightly higher bulk as well as grain boundary conductivity compared to the original composition BaCe0.9Y0.1O3-δ. On the other hand, the Ba0.95Sr0.05Ce0.9Y0.1O3 sample also exhibits improved electrical conductivity in hydrogen gas atmospheres or atmospheres involving humidity. The greater chemical resistance of Ba1-xSrxCe0.9Y0.1O3, where x 0.02; 0.05 in a CO2 gas atmosphere enhances the suitability of the modified material as an electrolyte for fuel cells with proton-conducting ceramics. The article is quite interesting; it includes enough experimental work and adds some new information. Furthermore, it adheres to the journal’s standards as Materials provides a forum for publishing papers that advance in-depth understandings of the relationships between the structures, properties, applications or functions of all classes of materials. Thus, my recommendation is to be accepted with minor revision.

Specific Comments:

- English should be improved by a native speaker

- The innovation of the presented work should be clarified

- Some more recent works from the literature that should be added and discussed through this manuscript.

  1. Sudhanshu Dwivedi, Solid oxide fuel cell: Materials for anode, cathode and electrolyte, i n t e r n a t i o n a l journal o f hydrogen energy xxx ( x x x x ) xxx
  2. Donglin Han, Akiko Kuramitsu, Takayuki Onishi, Yohei Noda, Masatoshi Majima, Tetsuya Uda, Fabrication of protonic ceramic fuel cells via infiltration with Ni nanoparticles: A new strategy to suppress NiO diffusion & increase open circuit voltage, Solid State Ionics 345 (2020) 115189
  3. Chaoliang Geng, Xinxi Yu, Piaopiao Wang, Jigui Cheng, Tao Hong, The rapid one-step fabrication of bilayer anode for protonic ceramic fuel cells by phase inversion tape casting, Journal of the European Ceramic Society xxx (xxxx) xxx–xxx
  4. Erdienzy Anggia, Eun-Kyung Shin, Jun-Tae Nam, Jong-Sung Park, Fabrication of ceramic composite anode at low temperature for high performance protonic ceramic fuel cells, Ceramics International 46 (2020) 236–242
  5. М.S. Plekhanov, A.V. Kuzmin, E.S. Tropin, D.A. Korolev, M.V. Ananyev, New mixed ionic and electronic conductors based on LaScO3: Protonic ceramic fuel cells electrodes, Journal of Power Sources 449 (2020) 227476

Author Response

Dear Reviewer 2

Thank you very much for reviewing of our paper. Your valuable remarks allowed us to improve the quality of our paper. The all answers for your remarks/suggestions were included in seperate file (Pdf file) attached here

Best regards

Magdalena Dudek

Reviewer 3 Report

This manuscript talks about partial replacement of Ba by Sr in BaCe(Y)O3 perovskite, targetting the application in CO2 containing hydrogen fuel cells. Overall it is well structured with clear flow of objectives and results. Although not much exciting discoveries, the results do show minor improvement of the ionic conductivity with a little incorporation of Sr, which also enhances the resistance against CO2.

If any improvement, it should be the "highlight of the major findings" out of this systematic investigation of the variable composition of Ba/Sr, viz., to point out to the readers explicitly, which has been sort of 'buried' so far.

Author Response

Dear Reviewer 3

Thank you very much for reviewing our paper entitled  The Ba1–xSrxCe0.9Y0.1O3, 0 < x < 0.1, with improved chemical stability  in CO2 gas involving atmospheres as potential electrolytes for proton ceramic fuel cell

 Your valuable remarks enabled us to improve the quality of our paper. All answers were collected in this table

The English language was corrected by native speaker in the revised version of manuscrpit

Best reagrds

Magdalena Dudek

Round 2

Reviewer 1 Report

The authors did not address the concerns in the manuscript raised in the previous revision. The authors responded to my questions in the cover letter but nothing has been added to the manuscript. The authors should include the discussion part related to the concerns raised earlier. For example, the authors responded in cover letter/response letter with XRD diffraction peak shift towards higher 2theta and it goes along with previous reports (I already suggested it) and so why not comment on this in the manuscript?

Author Response

Dear Reviewer 1,

Dear Editors,

Thank you very much for reviewing our paper.  Your valuable remarks enabled us to improve the quality of our paper. Now I would like to take this opportunity to respond to all your remarks

Remark 1

The authors did not address the concerns in the manuscript raised in the previous revision. The authors responded to my questions in the cover letter but nothing has been added to the manuscript. The authors should include the discussion part related to the concerns raised earlier. For example, the authors responded in cover letter/response letter with XRD diffraction peak shift towards higher 2theta and it goes along with previous reports (I already suggested it) and so why not comment on this in the manuscript?

Answer1

Yes, authors confirm that they provided the answers for all your sugestions

In previous version we did not include new images, coments and discusion in the manuscript, because we are affraid the whole manuscrpit will be to long and have two many figures. Now according to Your suggestions we attached the results in the text.  In the file RemarkRev1604 we have listen all changes, which we have made .

All your suggestions were included in the text

The whole manuscript was revised by English Native speaker

Round 3

Reviewer 1 Report

The authors addressed the concerns and recommend the manuscript to publish as is.